# Dimethyl Fumarate Suppresses Demyelination and Axonal Loss through Reduction in Pro-Inflammatory Macrophage-Induced Reactive Astrocytes and Complement C3 Deposition

**DOI:** 10.3390/jcm10040857

**Published:** 2021-02-19

**Authors:** Sudhir K. Yadav, Naoko Ito, Devika Soin, Kouichi Ito, Suhayl Dhib-Jalbut

**Affiliations:** 1Department of Neurology, Rutgers-Robert Wood Johnson Medical School, Piscataway, NJ 08854, USA; yadavsk@rwjms.rutgers.edu (S.K.Y.); itona@rwjms.rutgers.edu (N.I.); dsoin@student.touro.edu (D.S.); 2Touro College of Osteopathic Medicine, Middletown, NY 10940, USA; 3Department of Neurology, New Jersey Medical School, Newark, NJ 07101, USA

**Keywords:** dimethyl fumarate, experimental autoimmune encephalomyelitis, iNOS^+^ macrophage, Ym1^+^ macrophages, reactive C3^+^ astrocytes, complement C3, proinflammatory, immunomodulatory, demyelination, axonal loss, neuroprotection, multiple sclerosis

## Abstract

Dimethyl fumarate (DMF) is an oral agent for relapsing-remitting multiple sclerosis (RRMS). In this study, we investigated the therapeutic mechanism of DMF using experimental autoimmune encephalomyelitis (EAE). DMF treatment decreased the proliferation of T cells and the production of IL-17A and GM-CSF. DMF treatment also decreased the development and/or infiltration of macrophages in the central nervous system (CNS), and reduced the ratio of iNOS^+^ pro-inflammatory macrophage versus Ym1^+^ immunomodulatory macrophages. Furthermore, DMF treatment suppressed the deposition of complement C3 (C3) and development of reactive C3^+^ astrocytes. The decrease in iNOS^+^ macrophages, C3^+^astrocytes, and C3 deposition in the CNS resulted in the reduction in demyelination and axonal loss. This study suggests that the beneficial effects of DMF involve the suppression of iNOS^+^ pro-inflammatory macrophages, C3^+^ astrocytes, and deposition of C3 in the CNS.

## 1. Introduction 

Multiple sclerosis (MS) is a chronic demyelinating disease of the central nervous system (CNS). Infiltration of pathogenic immune cells such as Th1 and Th17 cells, B-cells, and macrophages into the CNS cause disease initiation and progression [1,2]. Among the FDA-approved disease modifying agents for relapsing-remitting multiple sclerosis (RRMS), dimethyl fumarate (DMF) is an oral drug with neuroprotective and immunomodulatory effects [3,4,5]. Although the mechanism of action of DMF therapy is not completely understood, its therapeutic effect involves both nuclear factor erythroid 2-related factor 2 (Nrf2) dependent [6] and independent pathways [7]. Nrf2 is a transcription factor, which maintains oxidant homeostasis inside the cell by inducing an antioxidant response element (ARE)-containing gene, hemoxygenase-1 (HO-1) [8]. DMF confers neuroprotection during neuroinflammation through the induction of the Nrf2 anti-oxidative pathway in glial cells and neurons [6]. Recent studies also showed that anti-inflammatory effects of DMF are observed in a wide range of immune cells, including B cells, natural killer (NK) cells, T cells, dendritic cells (DCs) and macrophages through multiple targets such as hydroxycarboxylic acid receptor 2 (HCA2), interleukin-1 receptor-associated kinase 4 (IRAK4), glyceraldehyde 3-phosphate dehydrogenase (GAPDH), and nuclear factor kappa-light-chain-enhancer of activated B cells (NF-κB) [9,10,11,12]. Therefore, DMF can suppress the development of pathogenic immune cells in the periphery, migration of pathogenic immune cells into the CNS, and induction of neuroinflammation through several different pathways.

Demyelination and axonal loss are hallmark characteristics of MS, and glia cells including microglia and astrocytes play a pivotal role in demyelination and axonal loss. In particular, the development of macrophages/microglia expressing nitric oxide synthase (iNOS) and reactive C3^+^ astrocytes in the CNS is highly involved in disease progression [13,14,15,16,17]. Proinflammatory cytokines produced by infiltrated immune cells and TLR ligands promote the development of iNOS^+^ macrophages and C3^+^ astrocytes [13,16,18]. Importantly, C3 produced in glia cells plays an important role in the development of pro-inflammatory macrophages and reactive astrocytes [13,19,20]. Ewe investigated the effect of DMF treatment on the development of iNOS^+^ pro-inflammatory macrophages, Ym-1^+^ immunomodulatory macrophages, reactive C3^+^ astrocytes, and C3 deposition in the CNS.

## 2. Materials and Methods

### 2.1. Induction of EAE and DMF Treatment

C57BL/6 mice were purchased from the Jackson Laboratory (Bar Harbor, ME) and housed in a specific pathogen-free facility at the Rutgers Robert Wood Johnson Medical School (Piscataway, NJ, USA). MOG-EAE; Experimental autoimmune encephalomyelitis (EAE) was induced by the subcutaneous immunization of 7-week-old C57BL/6 mice with 200 µL emulsions of 200 µg MOG35-55 peptide (MEVGWYRSPFSRVVHLYRNGK; Protein and Nucleic Acid Core Facility, Stanford University, Stanford, CA) in Complete Freund’s Adjuvant (CFA). Additionally, animals received an intraperitoneal injection of 100 µg pertussis toxin (List Biologicals, St. Louis, MO) on Day 0 and Day 2. Then, mice were orally administered with dimethyl fumarate (Biogen Idec, MA, USA) dissolved in 0.8% methocel at 100 mg/kg every day from one day post immunization. The control group received 0.8% methocel by oral gavage. Clinical signs of EAE were assessed according to the following scale: 0: no signs of disease; 1: limp tail; 1.5: paresis of one hindlimb; 2: paresis of both hindlimbs; 2.5: paralysis of one hindlimb; 3.0: paralysis of both hindlimbs; 3.5: paralysis of both hindlimbs and one forelimb paresis; 4: hindlimb paralysis and both forelimb paresis; 5: no mobility/moribund. All animal studies were conducted in accordance with Institutional Animal Care and Use Committee (IACUC) guidelines and were approved by Rutgers Robert Wood Johnson Medical School Animal Care and Use Committee (IACUC approval number: PROTO999900129).

### 2.2. ELISA and T Cell Proliferation Assay

Spleen and lymph node cells were cultured at 2 × 10^6^ cells/mL with anti-CD3 mAb (1 µg/mL) and anti-CD28 mAb (1 µg/mL), or MOG35-55 (10 µg/mL) at 37 °C for 72 h. The production of IFN-γ, IL-17A, GM-CSF and IL-10 was measured by ELISA (eBioscience, San Diego, CA, USA). To measure the proliferation of cells, [^3^H]-thymidine (1 µCi) was added to each well for 15 h, and cells were harvested and counted on a TriLux Liquid Scintillation Counter (Wallac, Boston, MA, USA). Proliferation was examined by [^3^H] thymidine uptake.

### 2.3. Immunohistology

Animals were anesthetized and perfused intracardially with 30 mL ice-cold PBS and 100 mL of 4% paraformaldehyde. Spinal cords were removed and soaked in 30% sucrose/PBS for 3 days. Frozen spinal cord sections (cervical, thoracic, lumbar, and sacral regions) were cut at a thickness of 20 µm by cryostat and stained with rabbit anti-myelin basic protein (MBP) and rat anti-neurofilament monoclonal antibodies (Abcam, Cambridge, MA, USA) to study demyelination and axonal loss, respectively. Rabbit anti-Ym-1 Ab (Abcam, Cambridge, MA, USA) was used to investigate immunomodulatory macrophages, while rabbit anti-iNOS Ab (Abcam) was used as a marker for pro-inflammatory macrophages. Rat anti-CD68 Ab (Biorad, Hercules, CA, USA) was used to detect macrophages. Mouse anti-GFAP-Alexa fluor 488 Ab (Biolegend, San Diego, CA, USA) and rabbit anti-C3 Ab (Novus Biologicals, Littleton, CO, USA) were used to investigate reactive C3^+^ astrocytes. Primary antibodies were visualized with the following fluorescent secondary antibodies; donkey anti-rabbit IgG Alexa 488, goat anti-Rat IgG Rhodamine, goat anti-rabbit IgG Rhodamine, and chicken anti-rat IgG Alexa488 (Jackson Immunoresearch Laboratories, Inc., West Grove, PA, USA). Cell nuclei were counterstained with DAPI. Digital images of spinal cord sections were taken with an Axiovert 200 (Zeiss, Jena, Germany) and Leica DMi8 (Leica, Wetzlar, Germany) microscope. Areas of demyelination and axonal loss, as well as the total area of spinal cord, were determined by Image-J software. The percentage of lesions was calculated using the following equation: (lesion area/total spinal cord area) ×100. The area of pro-inflammatory macrophage was identified as CD68^+^iNOS^+^ cells; immunomodulatory macrophages were identified as CD68^+^Ym-1^+^; reactive C3^+^ astrocytes were identified as GFAP^+^C3^+^ cells using the threshold color setting in Image-J software (NIH, Bethesda, MD, USA) to identify double positive cells. The development and/or infiltration of iNOS^+^ macrophages, Ym-1^+^ macrophages, and reactive C3^+^ astrocyte were determined by measuring the infiltrated area within the total spinal cord area (µm^2^ /mm^2^).

### 2.4. Flow Cytometric Analysis

Briefly, 1 × 10^6^ cells were washed twice with staining buffer (0.1% NaN3, 2%FCS in PBS) and stained with antibodies against F4/80, iNOS, and CD206 (eBioscience, San Diego, CA, USA) according to the manufacturer’s instructions. The stained samples were analyzed by a Gallios Flow Cytometer using Kaluza analysis software (Beckman Coulter, Inc., Brea, CA, USA).

### 2.5. Statistical Analyses

GraphPad Prism 7 Software (GraphPad Software, Inc., San Diego, CA, USA) and R version 4.0.2 was used for statistical analyses. EAE clinical score was evaluated by the non-parametric Mann–Whitney U test. Student’s t-test (unpaired) was used to assess the differences between the two groups. Spearman correlation test was used to investigate correlation. *p*-values <0.05 were considered statistically significant.

### 2.6. Materials

MOG35-55 peptide (MEVGWYRSPFSRVVHLYRNGK) was purchased from Protein and Nucleic Acid Core Facility Stanford University, Stanford, CA, USA. Pertussis toxin was purchased from List Biologicals, St. Louis, MO, USA. Methocel was purchased from Sigma Aldrich, St. Louis, MO, USA. Rabbit anti-MBP Ab, rat anti-neurofilament Ab, rabbit Anti-Ym-1 Ab, rabbit anti-iNOS Ab for immunohistology were purchased from Abcam, Cambridge, MA, USA. Rat anti-CD68 Ab was purchased from Biorad, Hercules, CA, USA. Mouse anti-GFAP-Alexa fluor 488 Ab was purchased from Biolegend, San Diego, CA, USA. Rat anti-C3 Ab was purchased from Novus Biologicals, Littleton, CO, USA.

Donkey anti-rabbit IgG Alexa 488, goat anti-rat IgG Rhodamine, goat anti-rabbit IgG Rhodamine, and chicken anti-rat IgG Alexa488 were purchased from Jackson Immunoresearch Laboratories, Inc., West Grove, PA, USA. Antibodies against F4/80, iNOS, and CD206 for flow cytometry and ELISA kits for IFN-γ, IL-17A, GM-CSF and IL-10 were purchased from eBioscience, San Diego, CA, USA.

## 3. Results

### 3.1. Effect of DMF on Cytokine Production

To examine the effects of DMF treatment on the development of EAE, we employed MOG35-55-induced EAE [21]. We treated the mice with DMF at 100 mg/kg on Day 1 post-immunization and continued this regimen until the early chronic stage (25 days post-immunization). As a control, a DMF-solvent (methocel) was orally administered. The prophylactic treatment with DMF delayed disease onset and reduced the clinical severity (Figure 1).

We investigated the effect of DMF treatment on the proliferation of T cells in the spleen and mesenteric lymph node (MLN). DMF pre-treatment reduced the proliferation of MOG35-55-specific T cells in the spleen and MLN (Figure 2). Pro-inflammatory cytokines such as IFN-γ, IL-17A, and GM-CSF play critical roles in the initiation and progression of EAE, while IL-10 produced by regulatory T cells and innate immune cells plays a pivotal role in disease suppression [22]. Therefore, we investigated the effect of DMF treatment on the production of IFN-γ, IL-17A, GM-CSF, and IL-10. We observed a trend toward suppression of these cytokines in the spleen by DMF treatment (Figure 3). Production of IL-17A and GM-CSF in MLN was also suppressed by DMF treatment; however, production of IFN-γ in MLN was not suppressed by DMF treatment. Interestingly, production of IL-10 in the MLN was increased by DMF treatment (Figure 3).

### 3.2. DMF Treatment Decreased Demyelination and Axonal Loss

Demyelination and axonal loss are histopathological hallmarks of EAE and MS [23]. Therefore, we examined the effect of DMF on demyelination and axonal loss. Demyelination was predominantly observed in white matter around the areas of cellular infiltration, and axonal loss was accompanied by demyelination (Figure 4a). Notably, demyelination and axonal loss were significantly suppressed by DMF treatment (Figure 4b,c).

### 3.3. DMF Treatment Decreased iNOS^+^ Macrophages and Ym1^+^ Macrophages in the CNS

Pro-inflammatory iNOS^+^ microglia/macrophages are found in MS lesion and are involved in demyelination and axonal loss [15,16,24]. Notably, iNOS is also expressed by other glial and immune cells and can promote demyelination and axonal loss [15]. Conversely, Ym1^+^ or CD206^+^ macrophages are immunomodulatory macrophages involved in neuroprotection [25,26,27]. Therefore, we examined the effect of DMF on the development of iNOS^+^ macrophages and CD206^+^ or Ym1^+^ macrophages in the spleen and spinal cord, respectively. Although we did not observe the effect of DMF treatment on the development of F4/80^+^ iNOS^+^ and F4/80^+^ CD206^+^ macrophages in the spleen (Figure 5), DMF treatment suppressed the development and/or infiltration of iNOS^+^ macrophages and Ym1^+^ macrophages in the spinal cord (Figure 6). A higher number of macrophages (CD68^+^ cells) was detected in the spinal cord of control mice, and many of them expressed the iNOS (Figure 6a,b). Enrichment of iNOS^+^ macrophages in the spinal cord was significantly reduced in DMF-treated mice compared to control mice (Figure 6a,b). We also observed that the number of CD68^+^ Ym-1^+^ macrophages was reduced in the DMF-treated mice compared to control mice (Figure 6c,d). Importantly, the ratio of iNOS^+^ macrophages versus Ym1^+^ macrophages was reduced in DMF-treated mice (Figure 6e).

### 3.4. DMF Treatment Decreased Deposition of C3 and Development of Reactive C3^+^ Astrocytes in the CNS

Deposition of C3 has been reported to play a pathogenic role in neurodegeneration [13,28]. Since C3 is induced by TLR4/9-signaling and proinflammatory cytokines such as IL-1β and TNF-α [13,29], we examined the effect of DMF treatment on the production of C3 in the CNS of EAE mice. Although deposition of C3 was detected in the CNS upon development of EAE, DMF treatment suppressed C3 deposition (Figure 7a,c). Importantly, C3 is induced in astrocytes through exposure to proinflammatory cytokines, TNF-α, IL-1α, and C1qa, produced by activated pro-inflammatory microglia and/or macrophages, and the reactive C3^+^ astrocytes are highly neurotoxic [13]. MOG-EAE is associated with expansion of pro-inflammatory A1 astrocytes in the spinal cord during the development of EAE, whereas development of anti-inflammatory A2 astrocytes is not affected by EAE [30]. Therefore, we examined the effect of DMF treatment on the development of reactive C3^+^ astrocytes in the spinal cord. We used GFAP as a marker for astrocyte and detected GFAP^+^C3^+^ cells as pro-inflammatory reactive astrocytes as shown previously [13]. Although there was no difference in the area of GFAP^+^ astrocytes between DMF treated and control groups (Figure 7a,b)**,** DMF treatment suppressed the development of reactive C3^+^ astrocytes during the development of EAE (Figure 7a,d). Notably, there was a trend toward an increase in reactive C3^+^ astrocytes in the mice having a higher population of iNOS^+^ macrophages (Figure 7e). These data suggest that suppression in the development of reactive C3^+^ astrocytes could be one of the important therapeutic mechanisms of DMF treatment.

## 4. Discussion

DMF is an oral disease-modifying treatment approved for RRMS patients [5]. Both Nrf2-dependent and independent pathways are suggested to mediate the beneficial actions of DMF treatment [6,7,31]. Several earlier reports have demonstrated the prophylactic efficacy of DMF treatment on the development of EAE [6,7]. We also observed the prophylactic effect of DMF on active EAE (Figure 1). Activation of myelin-specific T cells and production of pro-inflammatory cytokines are the primary events in disease initiation and relapse. DMF treatment decreased the proliferation of MOG-specific T cells (Figure 2) along with the suppression of IL-17A and GM-CSF production (Figure 3). We also observed an increase in production of IL-10 in the mesenteric lymph nodes (Figure 3). Orally administered DMF may be able to promote the development of IL-10 producing immune cells in the gut, which might migrate to the periphery and suppress the differentiation of pathogenic T cells [32]. Further experiments are needed to explore the effect of DMF on intestinal immune cells involved in the suppression of CNS autoimmunity.

It has been demonstrated that fumaric acid esters are effective in the suppression of EAE, possibly by reducing the infiltration of macrophages [33]. Pro-inflammatory macrophages/microglia are highly involved in neurodegeneration [34], while immunomodulatory macrophages/microglia are involved in neuroprotection [25]. Pro-inflammatory microglia/macrophages commonly express iNOS and are often present in the MS lesions [15,35]. Demyelination and a higher degree of axonal damage are associated with iNOS expression in macrophages during EAE [17]. Nitric oxide (NO) generated by the iNOS contributes to the pathogenesis of MS. NO and its reactive derivative peroxynitrite (ONOO-) are cytotoxic to oligodendrocytes and neurons by inhibiting the mitochondrial respiratory chain [36,37]. On the other hand, Ym1^+^ macrophages possess anti-inflammatory and neuroprotective functions [25,26,27]. Therefore, we studied the effect of DMF treatment on the development and/or infiltration of iNOS^+^ macrophages and Ym1^+^ macrophages. We showed that DMF treatment reduced both iNOS^+^ and Ym-1^+^ macrophages/microglia; however, the ratio of iNOS^+^ versus Ym-1^+^ macrophages/microglia in the spinal cord was reduced by DMF treatment (Figure 6). Monomethyl fumarate was reported to convert the pro-inflammatory phenotype to an anti-inflammatory one in a microglia cell line through activation of the HCAR2 and AMPK/Sirt pathways and subsequent inhibition of NF-κB pathway [31]. Therefore, DMF may be able to reduce the activation of macrophages/microglia through HCAR2 and AMPK/Sirt pathways.

The complement system is involved in the pathogenesis of both EAE and MS. Especially, a deposition of C3 has been observed in a majority of MS lesions collected from the MS patients [38,39,40,41]. C3 has been suggested to trigger antibody-dependent demyelination, synapse loss, and neurodegeneration during the development of MS [42,43,44]. Although C3 is produced in the liver and migrate into tissues, C3 is also induced in the tissues, including the CNS. Since C3 is induced by pathogen-associated molecular patterns (PAMPs), such as lipopolysaccharide (LPS), zymosan, and CpGs [45], and proinflammatory cytokines, such as IL1β and TNFα [13,46], DMF treatment may reduce the deposition of C3 by suppressing the production of proinflammatory cytokines and TLR signaling pathways [47]. C3 is also a marker for reactive astrocytes [13]. In this study, we showed that DMF treatment can suppress the development of C3^+^ GFAP^+^ astrocytes. Reactive C3^+^ GFAP^+^ astrocytes harbor highly neurotoxic properties and are involved in induction of axonal loss and suppression of remyelination [13,28,48]. Since C3^+^ GFAP^+^ astrocytes are induced by IL-1α, TNFα, IL6, and C1q produced by activated microglia [13,29], DMF treatment may suppress the development of reactive C3^+^ GFAP^+^ astrocytes by reducing the development and/or infiltration of iNOS^+^ microglia/macrophages through the activation of both Nrf2-dependent and independent pathways [31,47,49,50]. Interestingly, GFAP intensity is higher in DMF-treated mice (Figure 7a,b). Since pro-inflammatory C3^+^ astrocytes were barely detected in DMF-treated mice (Figure 7a,d), DMF may be able to activate other types of astrocytes including anti-inflammatory astrocytes. Since a certain population of astrocytes does not express GFAP [51,52], a further experiment needs to be performed to investigate the effect of DMF on activation or suppression of each subtype of astrocytes during EAE. In conclusion, our study suggests that suppression of pro-inflammatory iNOS^+^ macrophages and ratio of iNOS^e^ versus Ym-1^+^ macrophages/microglia, C3 deposition, and development of reactive C3^+^ astrocytes in the CNS by DMF treatment could contribute to the reduction in demyelination and axonal loss in MS.

## Figures and Tables

**Figure 1 jcm-10-00857-f001:**
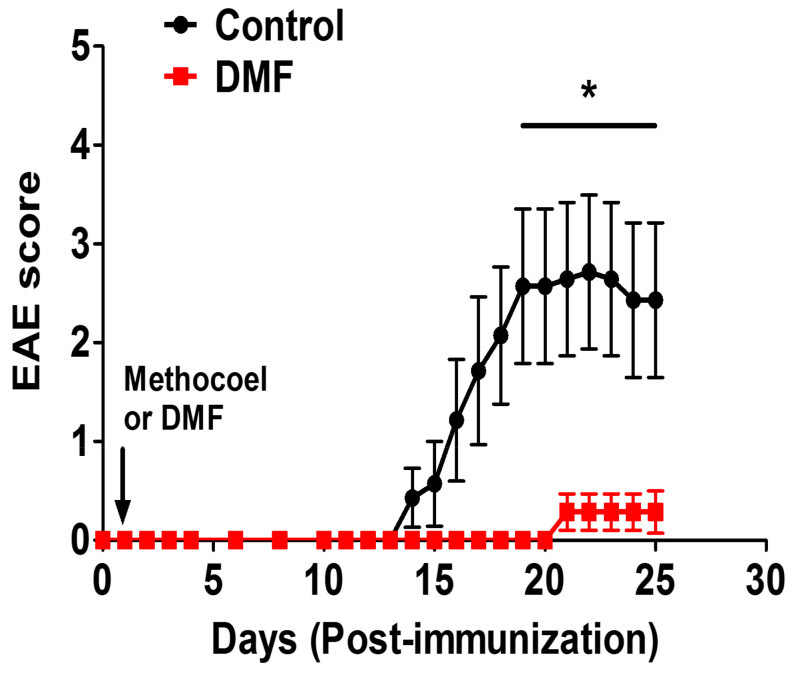
Effect of dimethyl fumarate (DMF) on MOG-EAE. EAE was induced in C57BL/6 mice by immunization with MOG35-55/CFA and injection of pertussis toxin, as described in Materials and Methods. The immunized mice were treated orally with DMF (*n* = 7) at 100 mg/kg or 0.8% methocel as a control (*n* = 7) every day from 1-day post-immunization until the end of the experiment. Data shown are the mean clinical score ± SEM. * *p*-value < 0.05 compared to a control.

**Figure 2 jcm-10-00857-f002:**
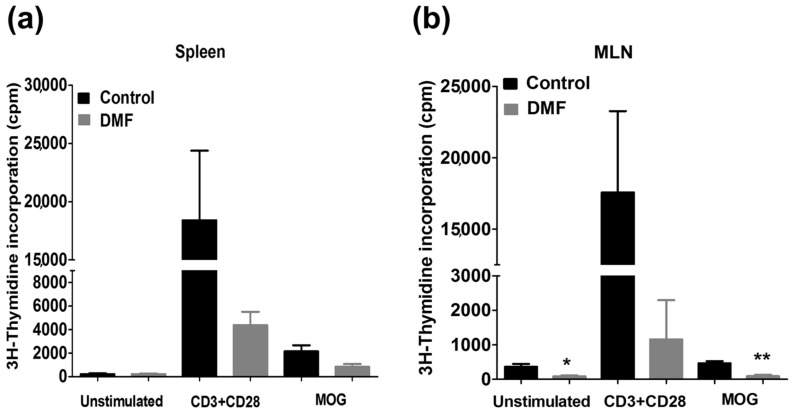
Effect of DMF on T cell proliferation in MOG-EAE mice. Mice were orally treated with DMF (100 mg/kg) or 0.8% methocel from day 1 to day 25 post-immunization. Cells isolated from the (**a**) spleen (*n* = 3–4) and (**b**) mesenteric lymph nodes (MLNs) (*n* = 3–4) were cultured with CD3 mAb and CD28mAb or MOG35-55 for three days and then examined for their proliferation by [^3^H]-thymidine uptake. * *p* < 0.05, ** *p* < 0.01.

**Figure 3 jcm-10-00857-f003:**
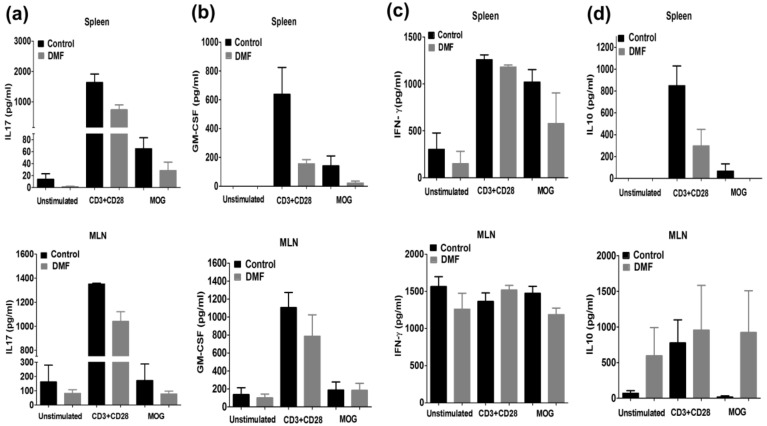
Effect of DMF on cytokine production in EAE mice. Mice were orally treated with DMF or 0.8% methocel from day 1 to day 25 post-immunization. The cells isolated from the spleen (*n* = 3–4) and MLNs (*n* = 3–4) were cultured with CD3 mAb and CD28 mAb or MOG35-55 for three days. Production of IL-17A (**a**), GM-CSF (**b**), IFN-γ (**c**), and IL-10 (**d**) was measured by ELISA.

**Figure 4 jcm-10-00857-f004:**
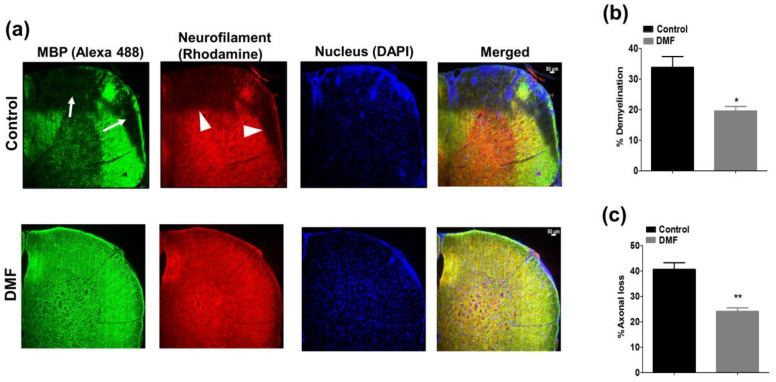
Effect of DMF treatment on demyelination and axonal loss. (**a**) Mice were euthanized 25 days after immunization, and demyelination and axonal loss were examined by staining with anti-myelin basic protein (MBP) and anti-neurofilament (NF) Abs, respectively. Cell nuclei were counterstained with DAPI (4’,6-diamidino-2-phenylindole). Areas of demyelination and axonal loss are shown by white arrows and arrow heads, respectively. Demyelination (**b**) and axonal loss (**c**) were estimated by measuring the MBP+, NF+, and total tissue areas as described in Materials and Methods. *n* = 3–4. * *p* < 0.05, ** *p* < 0.01 compared to control. Scale = 50 µm.

**Figure 5 jcm-10-00857-f005:**
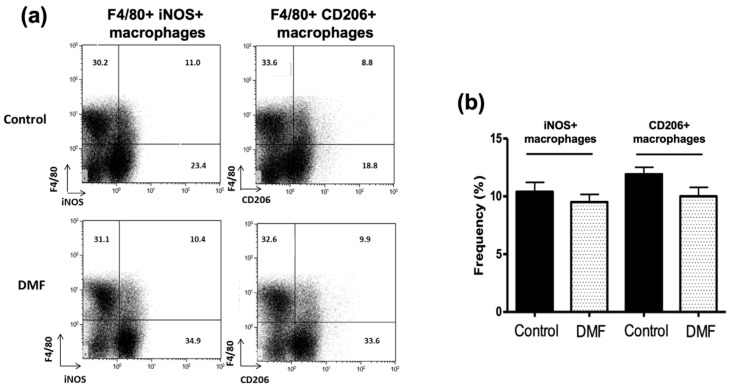
Effect of DMF treatment on development of macrophages in the spleen. (**a**) Development of pro-inflammatory (F4/80^+^iNOS^+^) and immunomodulatory (F4/80^+^CD206^+^) macrophages was examined by staining the splenocytes with F4/80, iNOS, and CD206 antibodies. Representative flow cytometry data are presented. (**b**) Quantification of iNOS^+^ and CD206^+^ macrophage population in the spleen are presented.

**Figure 6 jcm-10-00857-f006:**
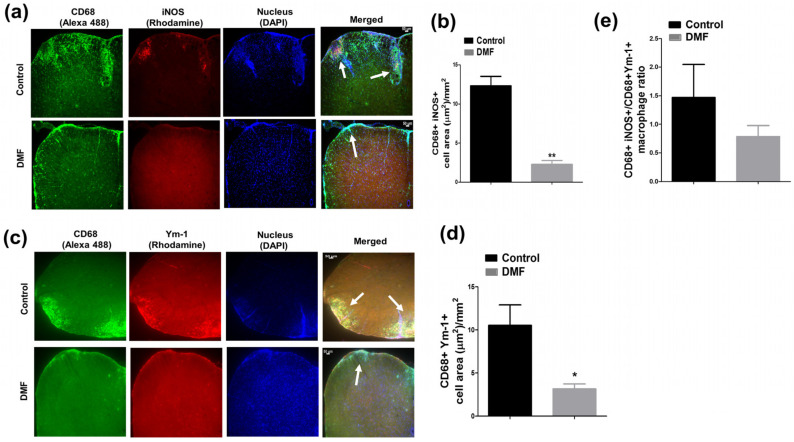
Effect of DMF treatment on development and/or infiltration of macrophages in the central nervous system (CNS). Mice were euthanized 25 days after immunization, and pro-inflammatory (CD68^+^iNOS^+^) and immunomodulatory (CD68^+^Ym-1^+^) macrophages in the spinal cord were examined by immunohistology. (**a**) Effect of DMF treatment on pro-inflammatory macrophages in the spinal cord. Pro-inflammatory macrophages were determined by co-staining with anti-iNOS and anti-CD68 Abs. Cell nuclei were counterstained with DAPI. An area of iNOS^+^ macrophage is shown by white arrows. (**b**) Area of iNOS^+^ macrophages was quantified as described in Materials and Methods. (**c**) Effect of DMF treatment on immunomodulatory macrophages in the spinal cord. Immunomodulatory macrophages were examined by co-staining with anti-Ym-1 and anti-CD68 Abs. Area of Ym-1^+^ macrophage is shown by white arrows. (**d**) Area of Ym-1^+^ macrophages was quantified as described in Materials and Methods. (**e**) Effect of DMF treatment on the ratio between iNOS^+^ and Ym-1^+^ macrophage in the spinal cord. *n* = 3–5. * *p* < 0.05, ** *p* < 0.01 compared to a control. Scale = 50 µm.

**Figure 7 jcm-10-00857-f007:**
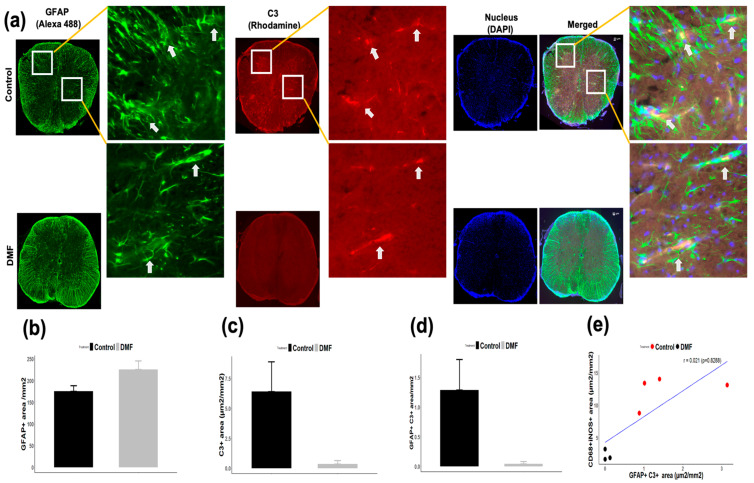
Effect of DMF treatment on deposition of C3 and development of reactive C3^+^ astrocytes in the CNS. (**a**) Mice were euthanized 25 days after immunization, and spinal cords were co-stained with GFAP and C3 mAbs. Cell nuclei were counterstained with DAPI. GFAP^+^ astrocytes (**b**), C3 (**c**) and GFAP^+^C3^+^ reactive astrocytes (**d**) were quantified as described in Materials and Methods. (**e**) The correlation between iNOS^+^ macrophages and reactive C3^+^ astrocytes is presented. Inset represents the magnified section of GFAP^+^ astrocytes, C3 deposit, and GFAP^+^C3^+^ reactive astrocytes. Scale = 50 µm.

## Data Availability

Not applicable.

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
