# Peer review of "Dimethyl Fumarate Suppresses Demyelination and Axonal Loss through Reduction in Pro-Inflammatory Macrophage-Induced Reactive Astrocytes and Complement C3 Deposition"

_jcm, 2021, doi:10.3390/jcm10040857_

Round 1

Reviewer 1 Report

Dimethyl Fumarate Suppresses Neurodegeneration Through Reduction of M1 Macrophages-Induced A1 Reactive Astrocytes and Complement C3 Deposition

This paper by Yadav et al is an investigation into the multiple sclerosis treatment dimethyl fumarate (DMF) and its possible mechanistic actions in reducing demyelination and axonal loss in an experimental model of multiple sclerosis- experimental autoimmune encephalomyelitis (EAE). By examining markers for the so called M1 (tissue destructive) and M2 ( cellular remodelling) macrophages/microglia, the so called A1 neurotoxic astrocytes, the C3 complement, and T cells they concluded that the DMF suppressed cytokines in the spleen, and DMF treatment decreased demyelination and axonal loss in the cord of the experimental animals. Also DMF treatment apparently decreased CNS infiltration of M1 and M2 macrophages and reduced the M1/M2 ratio. There was also evidence of decreased deposition of the complement component C3 and the development of A1 astrocytes in the CNS.

This is an interesting and well-presented study. However, I have a number of comments.

Major comments

1. The major drawback of this study is its apparent uncritical acceptance of the concept of both the M1/M2 macrophage theory and the A1/A2 astrocyte theory. There is certainly much scepticism as to the M1/M2 theory (Martinez and Gordon F1000 Prime Reports 2014: 6 ;13; Ransohoff Nat NeuroSci 2016: 19; 987-9991.). This should be at least discussed in the Discussion section. It is thought that this polarizing of M1 and M2 may just be an extreme induced by experimental conditions and bear little relationship to the true human disease process. There is also emerging scepticism of the A1 and A2 astrocyte model of disease originally suggested by Liddelow et al. Again there is a view that his polarizing of A1 and A2 may just be an extreme artificial experimental phenomena and may oversimplify the disease process. (Cunningham et al : Neuroscientist 2019: 25(5); 455-474).  There is recent work by Bayraktar et al (Nat Neurosci 2020: 23;500-509) that suggests that there is a diversity of astrocytic features across cerebral cortical layers and these differences  due to cues from local neurons rather than the astrocytes adopting one of 2 possible phenotypes. All this should be properly discussed in the discussion section.

2.Why were markers for A2 astrocytes not employed? One could argue that EAE and MS have more similarities to ischemia/infarction than to neurodegenerative diseases (such as Alzheimer’s disease and ALS) and ischemia/ infarction is supposed to produce predominantly A2 astrocytes according to Liddelow et al. The limitations of not searching for A2 astrocytes should be discussed.

3. Using only one marker for A1 astrocytes (C3) is a limitation and should be mentioned.

  1.  
  2. GFAP does not label all astrocytes and indeed Liddelow et al employed S100beta as an astrocyte marker for some of their experiments and GFAP for others. This may therefore skew the results when using C3/GFAP as a marker for A1 astrocytes. This should be mentioned in the text. 
  3.  
  4. It appears to me that this is an inappropriate journal for this paper. It is a purely experimental paper yet it is submitted to Journal of Clinical Medicine. Brain Sciences may be more appropriate.

Minor Comments:

  1. I take issue with “neurodegeneration” in the title . I would suggest demyelination in its place. MS and EAE are not truly primarily neurodegenerative diseases.

  1. Typographical error. Page 2 line 83 should be “lumbar” not “lumber”.

Author Response

Reviewer 1
Major comments:
1. The major drawback of this study is its apparent uncritical acceptance of the
concept of both the M1/M2 macrophage theory and the A1/A2 astrocyte theory.
There is certainly much scepticism as to the M1/M2 theory (Martinez and
Gordon F1000 Prime Reports 2014: 6 ;13; Ransohoff Nat NeuroSci 2016: 19; 987-
9991.). This should be at least discussed in the Discussion section. It is thought
that this polarizing of M1 and M2 may just be an extreme induced by experimental
conditions and bear little relationship to the true human disease process. There is
also emerging scepticism of the A1 and A2 astrocyte model of disease originally
suggested by Liddelow et al. Again there is a view that his polarizing of A1 and A2
may just be an extreme artificial experimental phenomena and may oversimplify
the disease process. (Cunningham et al : Neuroscientist 2019: 25(5); 455-
474). There is recent work by Bayraktar et al (Nat Neurosci 2020: 23;500-509) that
suggests that there is a diversity of astrocytic features across cerebral cortical
layers and these differences due to cues from local neurons rather than the
astrocytes adopting one of 2 possible phenotypes. All this should be properly
discussed in the discussion section.
Response: We agree with the reviewer that M1/M2 macrophages and A1/A2 astrocytes
dichotomy may be an oversimplification.
Pro-inflammatory microglia/macrophages commonly express iNOS and are often
present in the MS plaques [1,2]. Demyelination and a higher degree of axonal loss are
associated with iNOS expression in macrophages during EAE [3]. Nitric oxide (NO)
generated by the iNOS contribute to the pathogenesis of multiple sclerosis. NO and its
reactive derivative peroxynitrite (ONOO-) are cytotoxic to oligodendrocytes and neurons
in culture by inhibiting the mitochondrial respiratory chain [4,5]. On the other hand, Ym1+
or CD206+ macrophages are expressed by immunomodulatory macrophages, which are
involved in neuroprotection [6-8]. Therefore, we changed the M1 and M2 macrophages
to iNOS+ and Ym-1+ macrophages, respectively, throughout the manuscript.
Also, we changed the term A1 astrocytes to reactive C3+ astrocytes throughout the
manuscript because C3 is induced in astrocytes through exposure to proinflammatory
cytokines; TNF-α, IL-1α and C1qa, and the activated reactive C3+ astrocytes are highly
neurotoxic [9].
2.Why were markers for A2 astrocytes not employed? One could argue that EAE
and MS have more similarities to ischemia/infarction than to neurodegenerative
diseases (such as Alzheimer’s disease and ALS) and ischemia/ infarction is
supposed to produce predominantly A2 astrocytes according to Liddelow et al.
The limitations of not searching for A2 astrocytes should be discussed.
Response: MOG-EAE is associated with expansion of pro‐inflammatory and
neurotoxic astrocyte subpopulation (A1 astrocytes) during the development of EAE,
whereas A2 astrocyte subpopulation does not change [10]. Therefore, we did not
investigate the A2 astrocytes in this study. See line 240-243.
3. Using only one marker for A1 astrocytes (C3) is a limitation and should be
mentioned.
1. GFAP does not label all astrocytes and indeed Liddelow et al employed
S100beta as an astrocyte marker for some of their experiments and GFAP for
others. This may therefore skew the results when using C3/GFAP as a marker
for A1 astrocytes. This should be mentioned in the text.
Response: We agree with the reviewer about this comment. We added this
limitation in line 308-310.
2. It appears to me that this is an inappropriate journal for this paper. It is a
purely experimental paper yet it is submitted to Journal of Clinical Medicine.
Brain Sciences may be more appropriate.
Response: We had received in the past from JCM an invitation to submit a paper on
the therapeutic mechanism of another RRMS drug. Since DMF is a commonly used
oral medication for RRMS, we decided to submit this paper describing the
therapeutic mechanism of DMF. Since clinical studies using animal models to
understand mechanisms have been published in this journal, we were encouraged
to submit ours to JCM.
Minor Comments:
1. I take issue with “neurodegeneration” in the title . I would suggest
demyelination in its place. MS and EAE are not truly primarily
neurodegenerative diseases.
Response: we changed the title to “Dimethyl fumarate suppresses demyelination
and axonal loss through reduction of pro-inflammatory macrophages-induced
reactive astrocytes and complement C3 deposition”
2. Typographical error. Page 2 line 83 should be “lumbar” not “lumber”.
Response: we corrected the typo in line 87.
References:
1. Hill, K.E.; Zollinger, L.V.; Watt, H.E.; Carlson, N.G.; Rose, J.W. Inducible nitric
oxide synthase in chronic active multiple sclerosis plaques: distribution, cellular
expression and association with myelin damage. J Neuroimmunol 2004, 151,
171-179, doi:10.1016/j.jneuroim.2004.02.005.
2. Schuh, C.; Wimmer, I.; Hametner, S.; Haider, L.; Van Dam, A.M.; Liblau, R.S.;
Smith, K.J.; Probert, L.; Binder, C.J.; Bauer, J., et al. Oxidative tissue injury in
multiple sclerosis is only partly reflected in experimental disease models. Acta
Neuropathol 2014, 128, 247-266, doi:10.1007/s00401-014-1263-5.
3. Aboul-Enein, F.; Weiser, P.; Hoftberger, R.; Lassmann, H.; Bradl, M. Transient
axonal injury in the absence of demyelination: a correlate of clinical disease in
acute experimental autoimmune encephalomyelitis. Acta Neuropathol 2006, 111,
539-547, doi:10.1007/s00401-006-0047-y.
4. Giovannoni, G.; Heales, S.J.; Land, J.M.; Thompson, E.J. The potential role of
nitric oxide in multiple sclerosis. Mult Scler 1998, 4, 212-216,
doi:10.1177/135245859800400323.
5. Liu, J.S.; Zhao, M.L.; Brosnan, C.F.; Lee, S.C. Expression of inducible nitric
oxide synthase and nitrotyrosine in multiple sclerosis lesions. Am J Pathol 2001,
158, 2057-2066, doi:10.1016/S0002-9440(10)64677-9.
6. Miron, V.E.; Boyd, A.; Zhao, J.W.; Yuen, T.J.; Ruckh, J.M.; Shadrach, J.L.; van
Wijngaarden, P.; Wagers, A.J.; Williams, A.; Franklin, R.J.M., et al. M2 microglia
and macrophages drive oligodendrocyte differentiation during CNS
remyelination. Nat Neurosci 2013, 16, 1211-1218, doi:10.1038/nn.3469.
7. Raes, G.; Van den Bergh, R.; De Baetselier, P.; Ghassabeh, G.H.; Scotton, C.;
Locati, M.; Mantovani, A.; Sozzani, S. Arginase-1 and Ym1 are markers for
murine, but not human, alternatively activated myeloid cells. J Immunol 2005,
174, 6561; author reply 6561-6562, doi:10.4049/jimmunol.174.11.6561.
8. Raes, G.; De Baetselier, P.; Noel, W.; Beschin, A.; Brombacher, F.;
Hassanzadeh Gh, G. Differential expression of FIZZ1 and Ym1 in alternatively
versus classically activated macrophages. J Leukoc Biol 2002, 71, 597-602.
9. Liddelow, S.A.; Guttenplan, K.A.; Clarke, L.E.; Bennett, F.C.; Bohlen, C.J.;
Schirmer, L.; Bennett, M.L.; Munch, A.E.; Chung, W.S.; Peterson, T.C., et al.
Neurotoxic reactive astrocytes are induced by activated microglia. Nature 2017,
541, 481-487, doi:10.1038/nature21029.
10. Borggrewe, M.; Grit, C.; Vainchtein, I.D.; Brouwer, N.; Wesseling, E.M.; Laman,
J.D.; Eggen, B.J.L.; Kooistra, S.M.; Boddeke, E. Regionally diverse astrocyte
subtypes and their heterogeneous response to EAE. Glia 2020,
10.1002/glia.23954, doi:10.1002/glia.23954.

Reviewer 2 Report

The paper of Yadav et al. it's really interesting, well written and well exposed. Although DMF is an oral agent for relapsing-remitting multiple sclerosis, its multiple properties have yet to be fully investigated, such insights would benefit a wide range of diseases affecting the central nervous system.

Below are my minor comments, prior to acceptance into JCM.

Line 6-11: adjust the font size, affiliations and emails have two different sizes.

In the materials and methods section, although the origin of each material is well specified, it would be appropriate to insert a 2.6 Materials paragraph, listing each substance and/or compound used.

Given the many abbreviations used, it would be appropriate to add a paragraph that summarizes them all. This should be placed after the discussion, before author contributions, precisely after line 257.

Author Response

Reviewer 2
Line 6-11: adjust the font size, affiliations and emails have two different sizes.
Response: We adjusted the font size by following the template provided by JCM.
In the materials and methods section, although the origin of each material is well
specified, it would be appropriate to insert a 2.6 Materials paragraph, listing each
substance and/or compound used.
Response: We added 2.6 Materials in Materials and Methods.
Given the many abbreviations used, it would be appropriate to add a paragraph
that summarizes them all. This should be placed after the discussion, before
author contributions, precisely after line 257.
Response: We added a list of abbreviations. See line 321-331.

Reviewer 3 Report

The authors claim to show:

  1. DMF decreased T-cell proliferation
  2. DMF decreased production of IL17A and GM-CSF
  3. DMF decreased infiltration of macrophages into CNS
  4. DMF reduced ratio of M1 to M2 macrophages
  5. DMF suppressed C3 deposition
  6. DMF suppressed A1 astrocytosis development

Overall, my comments on the authors success to demonstrate each point:

  1. Shown in vitro. Authors might consider further analysis to corroborate T-cell numbers in vivo
  2. Shown in vitro. Sufficient for purpose of this paper
  3. The authors provide no definitive proof that macrophages are infiltrating the CNS
  4. Shown sufficiently
  5. Shown sufficiently
  6. I have a personal disagreement with the authors definition of A1 astrocytosis.

Major comments:

  • Authors repeatedly make reference to M1 and M2 macrophages as definitive states of immune response. This is widely accepted to be far too simplistic a definition of immune response, however the terminology can be useful to give perspective of overall changing patterns of immune response when considering large datasets. The authors here use only one marker to define each of M1 and M2 macrophages, making the use of the M1 and M2 terminology completely inappropriate.

  • T cell proliferation was measured in vitro by culture of cells (spleen and lymph node). Authors might consider sterelogical T-Cell counts in these tissues to compare the absolute numbers of T-Cells to support in vitro assays to improve scientific soundness but not essential

  • Figure 6 – authors make repeated reference to macrophage infiltration. It is entirely possible CNS resident macrophages or microglia are responsible for this staining and the authors show no evidence of CNS cell infiltration.

  • In methods, controls used in this study are not clearly defined

Minor comments:

  • iNOS used to define M1 macrophages – it should be noted that iNOS has also been shown to be expressed by other cells of the CNS, such as astrocytes. The authors overcome this by analysing only iNOS expressed by CD68+ cells, but this needs to be made explicit by the authors

  • Page 5, line 151; comment about demyelination and axonal loss around area of cell infiltration, I assume this is in reference to neuroanatomy? If so, please make this more explicit as one can read this macrophage infiltration which is not discussed in figure 4. If, however, this is in reference to macrophage infiltration (see above my disagreement with the evidence provided for macrophage infiltration) this needs to be made explicit

  • Figure 4. This would be well served with a control reference to a non-EAE healthy control. It is unclear if this is the way the analysis is performed as no reference to how controls are determined is included in Methods

  • Page 5, line 162: paper continues to make reference to M1 and M2 as definitive characterisations which is widely accepted to be primitive definitions without enough scope to define the true variety of immune responses that can be achieved by cells such as macrophages. For this paper, it would be better served discussing iNOS and Ym-1 and their specific influence on demyelination and axonal loss as these are the only ways in which the immune responses are defined.

  • Figure 6 – reference to a non-EAE healthy control would help authors convey how DMF compares to healthy CNS tissue

  • Figure 7 – To define A1 astrocytes, like the broad and varied range of macrophage responses, with expression of a single protein I would argue is far too simplistic to true biological nature of reactive astrocytosis. Authors might perhaps concentrate on discussing the known role of C3 in demyelination and axonal loss, which can include references to A1 but not using this as sole definition.

Author Response

Reviewer 3
Major comments:
Authors repeatedly make reference to M1 and M2 macrophages as definitive
states of immune response. This is widely accepted to be far too simplistic a
definition of immune response, however the terminology can be useful to give
perspective of overall changing patterns of immune response when considering
large datasets. The authors here use only one marker to define each of M1 and M2
macrophages, making the use of the M1 and M2 terminology completely
inappropriate.
Response: We agree with the reviewer that M1/M2 macrophages and A1/A2 astrocytes
are oversimplifications.
Pro-inflammatory microglia/macrophages commonly express iNOS and are often
present in the MS plaques [1,2]. Demyelination and a higher degree of axonal loss are
associated with iNOS expression in macrophages during EAE [3]. Nitric oxide (NO)
generated by the iNOS contribute to the pathogenesis of multiple sclerosis. NO and its
reactive derivative peroxynitrite (ONOO-) are cytotoxic to oligodendrocytes and neurons
in culture by inhibiting the mitochondrial respiratory chain [4,5]. On the other hand, Ym1+
or CD206+ macrophages are expressed by immunomodulatory macrophages, which are
involved in neuroprotection [6-8]. Therefore, we changed the M1 and M2 macrophages
to iNOS+ and Ym-1+ macrophages, respectively throughout the manuscript.
Also, we changed the term A1 astrocytes to reactive C3+ astrocytes throughout the
manuscript because C3 is induced in astrocytes through exposure to proinflammatory
cytokines; TNF-α, IL-1α and C1qa, and the activated reactive C3+ astrocytes are highly
neurotoxic [9].
T cell proliferation was measured in vitro by culture of cells (spleen and lymph
node). Authors might consider sterelogical T-Cell counts in these tissues to
compare the absolute numbers of T-Cells to support in vitro assays to improve
scientific soundness but not essential
Response: We used a conventional ex vivo T cell proliferation assay to examine the
effect of drugs on growth of T cells. We showed the result of this experiment. We
appreciate the reviewer’s suggestion to use sterelogical T-cell counts in future
experiments.
Figure 6 – authors make repeated reference to macrophage infiltration. It is
entirely possible CNS resident macrophages or microglia are responsible for this
staining and the authors show no evidence of CNS cell infiltration.
Response: We agree with the reviewer. Enrichment of macrophages in the CNS could
not be due to only infiltration. We changed “infiltration” to “development and/or
infiltration”. See line 17, 104,204,220, 282,303.
In methods, controls used in this study are not clearly defined
Response: We used 0.8% methocel to dissolve DMF, the mice treated with 0.8%
methocel were used as controls. See line 67.
Minor comments:
iNOS used to define M1 macrophages – it should be noted that iNOS has also
been shown to be expressed by other cells of the CNS, such as astrocytes. The
authors overcome this by analysing only iNOS expressed by CD68+ cells, but this
needs to be made explicit by the authors.
Response: We agree that iNOS can be expressed by other glial and immune cells. We
added this information. see line 198.
Page 5, line 151; comment about demyelination and axonal loss around area of
cell infiltration, I assume this is in reference to neuroanatomy? If so, please make
this more explicit as one can read this macrophage infiltration which is not
discussed in figure 4. If, however, this is in reference to macrophage infiltration
(see above my disagreement with the evidence provided for macrophage
infiltration) this needs to be made explicit.
Response: To be more explicit, we described that demyelination was predominantly
observed around the areas of cellular infiltration in the white matter. see line 182.
Figure 4. This would be well served with a control reference to a non-EAE healthy
control. It is unclear if this is the way the analysis is performed as no reference to
how controls are determined is included in Methods.
Response: We used 0.8% methocel to dissolve DMF, the mice treated with 0.8%
methocel were used as controls. See line 67.
Page 5, line 162: paper continues to make reference to M1 and M2 as definitive
characterisations which is widely accepted to be primitive definitions without
enough scope to define the true variety of immune responses that can be
achieved by cells such as macrophages. For this paper, it would be better served
discussing iNOS and Ym-1 and their specific influence on demyelination and
axonal loss as these are the only ways in which the immune responses are
defined.
Response: We agree with the reviewer that M1/M2 macrophages represent the two
extreme phenotypes of continuum and is a primitive definition. We changed the M1 and
M2 macrophages to iNOS+ and Ym-1+ macrophages, respectively, throughout the
manuscript.
References:
1. Hill, K.E.; Zollinger, L.V.; Watt, H.E.; Carlson, N.G.; Rose, J.W. Inducible nitric
oxide synthase in chronic active multiple sclerosis plaques: distribution, cellular
expression and association with myelin damage. J Neuroimmunol 2004, 151,
171-179, doi:10.1016/j.jneuroim.2004.02.005.
2. Schuh, C.; Wimmer, I.; Hametner, S.; Haider, L.; Van Dam, A.M.; Liblau, R.S.;
Smith, K.J.; Probert, L.; Binder, C.J.; Bauer, J., et al. Oxidative tissue injury in
multiple sclerosis is only partly reflected in experimental disease models. Acta
Neuropathol 2014, 128, 247-266, doi:10.1007/s00401-014-1263-5.
3. Aboul-Enein, F.; Weiser, P.; Hoftberger, R.; Lassmann, H.; Bradl, M. Transient
axonal injury in the absence of demyelination: a correlate of clinical disease in
acute experimental autoimmune encephalomyelitis. Acta Neuropathol 2006, 111,
539-547, doi:10.1007/s00401-006-0047-y.
4. Giovannoni, G.; Heales, S.J.; Land, J.M.; Thompson, E.J. The potential role of
nitric oxide in multiple sclerosis. Mult Scler 1998, 4, 212-216,
doi:10.1177/135245859800400323.
5. Liu, J.S.; Zhao, M.L.; Brosnan, C.F.; Lee, S.C. Expression of inducible nitric
oxide synthase and nitrotyrosine in multiple sclerosis lesions. Am J Pathol 2001,
158, 2057-2066, doi:10.1016/S0002-9440(10)64677-9.
6. Miron, V.E.; Boyd, A.; Zhao, J.W.; Yuen, T.J.; Ruckh, J.M.; Shadrach, J.L.; van
Wijngaarden, P.; Wagers, A.J.; Williams, A.; Franklin, R.J.M., et al. M2 microglia
and macrophages drive oligodendrocyte differentiation during CNS
remyelination. Nat Neurosci 2013, 16, 1211-1218, doi:10.1038/nn.3469.
7. Raes, G.; Van den Bergh, R.; De Baetselier, P.; Ghassabeh, G.H.; Scotton, C.;
Locati, M.; Mantovani, A.; Sozzani, S. Arginase-1 and Ym1 are markers for
murine, but not human, alternatively activated myeloid cells. J Immunol 2005,
174, 6561; author reply 6561-6562, doi:10.4049/jimmunol.174.11.6561.
8. Raes, G.; De Baetselier, P.; Noel, W.; Beschin, A.; Brombacher, F.;
Hassanzadeh Gh, G. Differential expression of FIZZ1 and Ym1 in alternatively
versus classically activated macrophages. J Leukoc Biol 2002, 71, 597-602.
9. Liddelow, S.A.; Guttenplan, K.A.; Clarke, L.E.; Bennett, F.C.; Bohlen, C.J.;
Schirmer, L.; Bennett, M.L.; Munch, A.E.; Chung, W.S.; Peterson, T.C., et al.
Neurotoxic reactive astrocytes are induced by activated microglia. Nature 2017,
541, 481-487, doi:10.1038/nature21029.

Reviewer 4 Report

The group of  Yadav  et al. investigated the anti-inflammatory effects  of DMF in a mouse model of  experimental   autoimmune encephalomyelitis. The authors show that DMF treatment reduced  pro-inflammatory responses and leaded to reduced demielynation and neurite loss. The authors have presented the results clearly, However, the paper needs some improvements   before it can be considered for publication.

  • In materials and methods the type (mouse, rat rabbit) of myelin basic protein (MBP) and neurofilament monoclonal antibodies (Abcam) are not indicated.
  • 3: what happens in cells obtained from non treated animals (non EAE) following stimulation? are  comparable to cells obtained from DMF animals? this result is important, because fig 4 shows in DMF animals no demielinization and neurite loss at all.
  • in fig.5 caption M2 (F4/80+iNOS+) should be changed to M2 (F4/80+206)
  • in 3.4, is the r value of the correlation statistically significant?
  • In fig 7a pictures of the inlets are not clear, higher magnification is needed in order to see colocalisation. In DMF treated mice GFAP immunofluorescence intensity is higher compared to control, suggesting that  that there is gfap overexpression, typical of reactive astrogliosis. Authors should discuss this result.
  • In fig. 7b-c graph of gfap+ area (only) quantification should be added; besides, in fig 7c how have been done quantification? how have been selected double staining? from images, seems very difficult to differentiate  double stained from gfap-only stained cells.

Author Response

Reviewer 4
In materials and methods the type (mouse, rat rabbit) of myelin basic protein
(MBP) and neurofilament monoclonal antibodies (Abcam) are not indicated.
Response: Rabbit anti-myelin basic protein and rat anti-neurofilament monoclonal
antibodies (Abcam, Cambridge, MA) were used to study demyelination and axonal loss,
respectively. We described this information in Materials and Methods. See in line 88-89.
3: what happens in cells obtained from non treated animals (non EAE) following
stimulation? are comparable to cells obtained from DMF animals? this result is
important, because fig 4 shows in DMF animals no demielinization and neurite
loss at all.
Response: We do not have data on naïve animals (non EAE) to compare with DMF
treated mice. Demyelination and axonal loss could be very mild because average EAE
score in DMF-treated mice is less than 0.5 in Fig. 1.
in fig.5 caption M2 (F4/80+iNOS+) should be changed to M2 (F4/80+206)
Response: We made correction for typo in figure 5 caption.
in 3.4, is the r value of the correlation statistically significant?
Response: r values is not statistically significant. However, there was a trend toward an
increase in reactive C3+ astrocytes in the mice having a higher population of iNOS+
macrophages. See line 247-248.
In fig 7a pictures of the inlets are not clear, higher magnification is needed in
order to see colocalisation. In DMF treated mice GFAP immunofluorescence
intensity is higher compared to control, suggesting that that there is gfap
overexpression, typical of reactive astrogliosis. Authors should discuss this
result.
Response: We magnified the merged picture. Although the reason why GFAP intensity
is higher in DMF-treated mice is uncertain, DMF may be able to activate certain types of
astrocytes. Since pro-inflammatory C3+ astrocytes were barely detected in DMF-treated
mice, DMF may be able to activate other types of astrocytes including anti-inflammatory
astrocytes. Future experiments are required to explore this possibility. See line 305-308.
In fig. 7b-c graph of gfap+ area (only) quantification should be added; besides, in
fig 7c how have been done quantification? how have been selected double
staining? from images, seems very difficult to differentiate double stained from
gfap-only stained cells.
Response: We added the quantification of GFAP+ area in figure 7b.
We used threshold color setting in image J software (NIH, Bethesda, MD) to identify
double positive cells. See line 101-106.

Round 2

Reviewer 4 Report

The authors have addressed the major  points highlighted in my revision by adding magnified inlets and quantification of GFAP fluorescence, as well as the other ones by punctual responses. It’s a pity that there are no  data on cells from non EAE animals, because the comparison with the results on DMF treated mice  could have emphasized  the beneficial effects of  DMF. Anyway The work can be accepted in the present form.